# Gold Nanoparticles for Wound Healing in Animal Models

**DOI:** 10.3390/nano15161213

**Published:** 2025-08-08

**Authors:** Stephen Klavsen, Sten Rasmussen

**Affiliations:** Department of Clinical Medicine, Aalborg University, 9260 Aalborg, Denmark

**Keywords:** gold nanoparticles, wound healing, full-thickness excisional wounds, preclinical animal models, nanoparticle therapy, systematic review, meta-analysis

## Abstract

Background: Gold nanoparticles (GNPs) are increasingly studied for their potential to enhance wound healing, but their overall efficacy remains uncertain. Methods: We conducted a systematic meta-analysis (search date: 14 May 2025) across five databases. Included were randomized animal studies comparing GNPs to placebo, reporting wound closure percentages and relevant variance measures. Risk of bias was assessed using Cochrane and CAMARADES tools. Cohen’s d was used to estimate effect size under a random-effects model. Results: Thirty-one studies met the inclusion criteria. The pooled effect size was d = 4.52 (95% CI: 3.61 to 5.43; z = 9.73; *p* < 0.001), indicating a significant benefit of GNPs. Although heterogeneity was moderate to high, results consistently favored GNPs. Conclusion: GNPs significantly accelerate wound healing in animal models, supporting their potential as therapeutic agents.

## 1. Introduction

Wound healing is a complex biological process involving inflammation, tissue proliferation, and remodeling, all of which are essential to restoring skin integrity and function [1,2]. Despite advances in wound care technologies, effective treatment of chronic and acute wounds remains a major clinical challenge [3]. These wounds, particularly diabetic ulcers, pressure injuries, and venous leg ulcers, are associated with significant morbidity, high recurrence rates, escalating healthcare costs and have a notable impact on patient quality [4]. The global burden of non-healing wounds continues to grow, driven by aging population and comorbidities [3,4].

To address the limitations of conventional wound care, increasing attention has turned to nanotechnology-based therapies [5]. Gold nanoparticles (GNPs) have received particular attention as delivery systems for therapeutic agents. GNPs have emerged as a promising therapeutic option due to their distinct physicochemical properties and biological activity, proving effective across different models and applications [5,6]. Preclinical animal studies have reported wound healing improvements compared to controls [7,8,9]. However, the magnitude and consistency of these effects remain unclear, as no prior study has systematically compared wound healing outcomes across preclinical models.

Chronic wounds, or non-healing wounds, present a significant global health challenge, primarily due to infections caused by bacteria and fungi [4,10]. Among these, diabetic foot and leg ulcers are particularly burdensome, with a prevalence estimated at 2.21 per 1000 people [3,4]. Diabetic ulcers are becoming increasingly common as the population ages and diabetes prevalence increases. These ulcers lead to significant patient disability, higher mortality rates, and considerable economic costs [4,11]. Factors contributing to their development include neuropathy, limited range of motion, pressure, ischemia, and infection. Approximately 15% of individuals with diabetes may develop an ulcer at some point in their lives, with a high likelihood of recurrence. Additionally, more than half of foot ulcers become infected, and infections are the leading cause of lower extremity amputations. Despite all the available diagnostic and therapeutic tools, the treatment for diabetic foot ulcers is the rate and progression of healing [12].

Pressure injuries, also known as pressure ulcers, are localized damage to skin and underlying tissue, typically over a bony prominence. These injuries result from prolonged pressure or a combination of pressure and shear forces. Risk factors include advanced age, malnutrition, and certain medical conditions [13]. Pressure injuries represent a global challenge in healthcare settings, particularly in hospitals [13,14]. The pooled estimate of pressure injury incidence is approximately 12%, with Europe reporting the highest prevalence and incidence rates [13,15]. Venous leg ulcers are open wounds on the lower leg, typically between the ankle and mid-calf, that persist for over six weeks. They are caused by venous blood flow, which leads to increased venous pressure in the attached area [16]. Venous ulcers make up approximately 70% of chronic leg ulcers, leading to a significant economic burden due to lengthy treatment times and high recurrence rates. While diagnostic and therapeutic options exist, slow or stalled healing remains the primary clinical obstacle [16,17].

GNPs support wound healing through multiple mechanisms. They have been shown to downregulate transforming growth factors, inflammatory cytokines, and angiogenesis markers [18,19]. They also exhibit antibacterial and antioxidant properties [20,21], which are essential for promoting wound healing and preventing wound infection [6]. Beyond wound care, recent clinical studies indicate that gold microparticles show promise in providing pain relief and modulating inflammation in the management of knee- and hip osteoarthritis [22,23,24,25]. These gold nano- or microparticles appear to influence cellular signaling pathways involved in tissue repair and regeneration [24,26,27,28,29,30]. Additionally, gold nanoparticles have been shown to enhance collagen formation and organization, leading to improved wound closure and less scarring [31,32].

The therapeutic effect of GNPs is strongly influenced by particle size and shape [32,33]. The most common shapes include spherical, rod-shaped, and caged nanoparticles. Spherical GNPs are known for their excellent biocompatibility and cellular uptake, while rod-shaped GNPs provide superior tissue penetration [34,35]. Caged GNPs have a higher drug-loading capacity, making them advantageous for targeted delivery, particularly in photothermal ablation compared to larger GNPs [35]. Triangular GNPs exhibit strong antibacterial properties, and star-shaped GNPs demonstrate high photothermal conversion efficiency [36,37]. Delivery platforms include hydrogels, nanofibers, microneedle patches, and sprays, each tailored to different wound types and treatment goals. The choice of delivery mechanism is determined by the wound’s characteristics and the desired therapeutic outcomes [38,39,40,41,42,43,44,45].

This review focuses on full-thickness excisional wounds, the most used preclinical model for studying impaired wound healing. These wounds are reproducible, easy to standardize, and closely mimic human wound healing, making them clinically relevant [2]. These wounds involve loss of both dermis and epidermis and pose major clinical risks due to barrier disruption [46]. In contrast, superficial wounds affect only the epidermis and are less clinically relevant in chronic wound research [47]. In current clinical practice for addressing severe burns and chronic wounds, surgical intervention is often recommended as a primary treatment strategy [48].

Despite encouraging findings, most studies remain limited to in vitro data or short-term in vivo trials, and no systematic meta-analysis has yet quantified GNP efficacy across animal wound models [6]. To our knowledge, there has been no systematic meta-analysis comparing the clinical efficacy of gold nanoparticle therapy on wound healing in randomized controlled trials involving animals. This systematic review and meta-analysis aimed to address the question: “What is the overall effect of gold nanoparticles on wound healing in animal models?” We also discuss the implications for clinical translation and future research. We hypothesized that gold nanoparticles significantly enhance the wound healing process in animal models by promoting healing.

## 2. Materials and Methods

### 2.1. Reporting and Registration

This report follows the PRISMA guidelines [49]. The protocol was registered in PROSPERO, the international register of systematic reviews (registration number: 622000), on 3 December 2024, and approved 9 April 2025 (PROSPERO 2025 CRD420250622000). The completed PRISMA checklist is provided in the Appendix A.

### 2.2. Information Sources

This study identified randomized controlled trials that investigated the effect of gold nanoparticles compared to a placebo control in the treatment of wound healing in animals. A systematic search was performed in the following databases: PubMed (MEDLINE), Embase, Cochrane, Scopus and Web of Science, without restriction on language or date of publication. 

### 2.3. Search Strategy

Three categories were used: wound healing, wound closure and gold nanoparticle treatment. The search was conducted by SK and SR, on 14 May 2025. Detailed search strategies for each database are presented in Appendix B (Table A1).

### 2.4. Study Selection

Duplicate entries were manually identified and removed by two reviewers (SK and SR) and independently performed the screening of eligible studies in two steps based on the PRISMA guidelines [49]. In the first step, all titles and abstracts were screened according to the predefined inclusion and exclusion criteria. The inclusion criteria consisted of animal randomized controlled studies that compared the effects of gold nanoparticle administration on wound healing outcomes to a placebo, with quantifiable data (SD or SEM) required for each group. In case of conflicts, both reviewers (SK and SR) re-evaluated the title and abstract and came to a mutual consensus after discussion.

In the second step, the two reviewers individually read all full-text articles included in the previous step. In addition to the formerly mentioned inclusion criteria, the final inclusion criteria were animal studies investigating the effects of gold nanoparticle administration on wound healing, compared to a placebo, specifically for measuring wound closure. Only studies reporting outcomes after three days were included to ensure that the initial inflammatory response had resolved and excluded after a maximum of 15 days to allow for a more focused evaluation of the early to mid-stages of wound healing in the proliferative phase [12,50,51]. Reasons for exclusion were pre-specified as the following: (1) no in vivo experiment, (2) no extractable or available data, (3) no placebo or control, (4) incorrect randomization or wrong study design, (5) not gold nanoparticles and (6) use of non-excisional wound models (e.g., burn, incisional, pressure ulcers, or chemically induced wounds). To reduce heterogeneity and ensure comparability, only studies using full-thickness excisional wound models were included. Other types of wounds (e.g., incisional, partial-thickness, ischemic, or infected wounds) were excluded. For statistical purposes all data were converted into means and standard deviations when possible. If this was not possible, the study was excluded due to available data.

### 2.5. Quality Assessment

The quality of the included articles was assessed for bias by the Cochrane Collaboration’s tool [52,53], which includes an assessment of bias in: Random sequence generation, allocation concealment, blinding of participants and personnel, blinding of outcome assessment, incomplete outcome data, selective reporting, and other biases. The quality assessment was conducted by both reviewers (SK and SR) and any disagreements were resolved by discussion. 

The Collaborative Approach to Meta-Analysis and Review of Animal Data from Experimental Studies (CAMARADES) checklist was used to evaluate the methodological quality of each individual study [54]. The tool includes 14 questions designed to identify potential biases in the study design. Scores range from 0 to 14, with higher scores indicating greater methodological quality [54,55].

### 2.6. Data Extraction

For each eligible study the following study characteristics were retrieved; year of publication, number of participants, type of gold nanoparticle used, type of control used, effect of gold nanoparticle, wound closure percentage and more than 3 days of treatment. If essential data were missing from the original article, the corresponding author was contacted. If there were more than two treatment groups in one study, only data from the relevant groups were extracted. In studies where data was not clearly stated, SK used a ruler to directly measure wound closure percentages from available images or diagrams. Wound closure was measured with a ruler in ten studies [56,57,58,59,60,61,62,63]. In studies where the wound closure percentage was not calculated, a formula was applied to convert wound healing data from absolute wound areas to percentages. The following formula was used to calculate percentage wound closure: (w_0_ − wₜ)/(w_0_ × 100). We extracted wound closure data from the time point where the gold nanoparticle intervention showed the largest effect compared to placebo.

### 2.7. Statistical Analysis

To perform the analyses the statistical software Stata/MP 18.0 was used. Statistical heterogeneity was tested by including Tau-squared (τ^2^), H-squared (H^2^), and I-squared (I^2^). This allows for a more comprehensive understanding of the variability among the results, helping to determine whether differences in study outcomes are likely due to true differences in study variables, rather than random variability. Pooled standard mean difference (SMD) with 95% confidence interval (CI) was calculated.

### 2.8. Data Presentation

Results of individual studies were summarized in structured tables or figures. Forest plots were generated for the meta-analysis to visually display effect sizes and confidence intervals. Heterogeneity was assessed and visually interpreted using these plots. Funnel plots were used to assess publication bias.

## 3. Results

### 3.1. Study Selection

A total of 529 records were identified through electronic databases, with an additional 20 studies found through citation searching and personal archives. After removal of duplicates and title/abstract screening, 146 full-text articles were assessed for eligibility. Ultimately, 31 studies met all inclusion criteria and were included in the final meta-analysis (Figure 1). The most frequent reasons for exclusion were the use of non-excisional wound models, wrong interventions, or lack of extractable outcome data. Only animal studies comparing gold nanoparticles to a placebo or control group and reporting quantitative wound closure outcomes were eligible.

### 3.2. Study Characteristics

A total of 31 studies was included, all of which investigated the effects of GNPs on full-thickness excisional wound healing in small animal models (mice, rats, rabbits). The studies spanned from 2012 to 2025, with sample sizes per group ranging from 3 to 18 animals. Rats were the most used species (20 studies), followed by mice (10 studies) and rabbits (1 study). Some studies used diabetic animal models, induced local infection, or applied antibacterial therapies, but in all cases, the primary wound type remained full-thickness excision. Table 1 presents the included study characteristics. The peak therapeutic effect, defined as the time point showing the greatest difference in wound closure between GNP-treated groups and controls, was most commonly observed between day 7 and day 15 post-intervention. This variation likely reflects differences in GNP formulation, delivery method, dosage, and experimental model. Study quality was assessed using a modified CAMARADES checklist, with scores ranging from 5 to 10 out of 14.

A wide range of GNP formulations and delivery strategies were employed. One study utilized a wound dressing and gel incorporating gold nanoparticles [20]. In four studies, multifunctional wound dressings were developed, enhancing antimicrobial activity [63,64,65,66]. Herbal gold nanoparticle therapy was explored in four studies [67,68,69,70]. Topical nanocomposite ointment therapy was employed in one study [71], while a multicomponent electrospun nanofiber wound dressing therapy was introduced in another [62]. A topical nanoantibiotic wound dressing therapy was explored in one study [60]. Antibacterial therapies were investigated in four studies [61,71,72,73] and Keratinocyte Growth Factor (KGF) conjugated with gold nanoparticles were utilized in two studies [58,74]. Photobiomodulation therapy with gold nanoparticles was examined in one study [7]. IL-4 modified gold nanozyme therapy was explored in two studies [75,76]. Nanocomposite wound healing therapy was emphasized for its application in diabetic foot ulcers in two studies [59,77]. Antioxidant gold nanoparticles were examined in two studies [78,79], while photothermal and oxidative therapy with nanorods was investigated in four studies [80,81,82,83]. Antimicrobial peptide-gold nanoparticle therapy was utilized in two studies [57,77]. Lastly, the topical application of green synthesized gold nanoparticles was addressed in four studies [56,69,70,84].

**Table 1 nanomaterials-15-01213-t001:** Study characteristics of the included studies.

Study	Intervention Group, n	Control Group, n	Wound Type	Animal Model	Day of Peak Effect	Study Quality
1. Chao et al., 2025 [76]	5	5	Full thickness	Mice	Day 7	7/14
2. Gerile et al., 2025 [82]	5	5	Full thickness	Mice	Day 7	7/14
3. Ma et al., 2025 [77]	5	5	Full thickness	Mice	Day 7	6/14
4. Biswal et al., 2025 [70]	3	3	Full thickness	Rats	Day 7	6/14
5. Cai et al., 2025 [69]	4	4	Full thickness	Rats	Day 8	8/14
6. Du et al., 2025 [65]	4	4	Full thickness	Mice	Day 9	7/14
7. Zhou et al., 2025 [73]	4	4	Full thickness	Mice	Day 3	5/14
8. Jiang et al., 2025 [83]	3	3	Full thickness	Mice	Day 13	7/14
9. Luo et al., 2025 [66]	3	3	Full thickness	Mice	Day 12	7/14
10. Salama et al., 2024 [84]	6	6	Full thickness	Rats	Day 9	7/14
11. Peng et al., 2024 [71]	6	6	Full thickness	Rats	Day 12	7/14
12. Jiang et al., 2024 [79]	5	5	Full thickness	Mice	Day 12	10/14
13. Li et al., 2023 [81]	3	3	Full thickness	Mice	Day 14	10/14
14. Yao et al., 2023 [75]	6	6	Full thickness	Mice	Day 15	7/14
15. Chen et al., 2023 [20]	6	6	Full thickness	Rats	Day 13	8/14
16. Wang et al., 2022 [64]	7	7	Full thickness	Rats	Day 7	10/14
17. Azlan et al., 2021 [63]	8	8	Full thickness	Rats	Day 3	8/14
18. Gharehpapagh et al., 2021 [67]	18	18	Full thickness	Mice	Day 7	9/14
19. Wang et al., 2020 [60]	12	12	Full thickness	Rats	Day 10	10/14
20. Al-Musawi et al., 2020 [62]	5	5	Full thickness	Rabbits	Day 7	7/14
21. Hu et al., 2020 [61]	5	5	Full thickness	Mice	Day 4	7/14
22. Li et al., 2019 [74]	6	6	Full thickness	Rats	Day 3	9/14
23. Sun et al., 2019 [72]	15	15	Full thickness	Mice	Day 7	7/14
24. Martínez et al., 2019 [59]	12	12	Full thickness	Mice	Day 10	10/14
25. Xu et al., 2018 [80]	3	3	Full thickness	Rats	Day 10	10/14
26. Pan et al., 2018 [58]	6	6	Full thickness	Rats	Day 10	10/14
27. Raghuwanshi et al., 2017 [68]	6	6	Full thickness	Rats	Day 9	10/14
28. Comune et al., 2017 [57]	10	10	Full thickness	Mice	Day 7	10/14
29. Lau et al., 2017 [7]	5	5	Full thickness	Rats	Day 7	9/14
30. Naraginti et al., 2016 [56]	6	6	Full thickness	Rats	Day 9	10/14
31. Leu et al., 2012 [78]	6	6	Full thickness	Mice	Day 7	10/14

Only studies reporting outcomes after three days were included to ensure that the initial inflammatory response had resolved [11,85,86,87]. This provided a clearer understanding of how the proliferative phase began without interference from acute inflammation. Measuring after this phase offers more reliable assessments of wound healing progression and treatment effects, by reducing variability in inflammatory responses [11,86]. Several studies observed peak differences at day 7, indicating a common timeframe where the therapeutic effects of gold nanoparticles are most pronounced [7,40,57,62,67,70,72,76,77,78,82]. Four studies noted peak differences on day 9 [56,65,68,84], while another four studies observed it at day 10 [58,59,60,80]. Three studies found the earliest peak effect at day 3 [63,73,74] and 7 studies found the latest peak effect between day 12–15 [20,55,66,71,75,81,83].

### 3.3. Study Quality and Publication Bias

All 31 studies were quality assessed using the Cochrane Collaboration’s tool for assessing risk of bias [53] (Table 2). All studies adequately described random sequence generation and were rated as low risk for this domain. However, allocation concealment was unclear in all studies, likely due to insufficient methodological reporting, which may introduce selection bias.

Blinding of investigators and outcome assessment was uniformly rated as high risk across all studies, reflecting common limitations in animal research where blinding procedures are often not implemented or reported. This raises concerns about performance and detection bias, particularly given the subjective nature of wound assessment.

Reporting bias and attrition bias were generally low, as most studies reported complete outcome data and did not show signs of selective reporting. No studies were identified as having other significant sources of bias.

While the overall methodological quality was moderate, the consistently high risk of bias in blinding and assessment underscores the need for more rigorously designed animal studies. This is particularly important given the visual nature of wound healing outcomes, which may be prone to observer expectation effects.

The overall quality score based on the CAMARADES tool ranged from 5 to 10, out of 14 with a median score of 8 (Table 3). All included studies were published in peer-reviewed journals using mice, rats or rabbits as animal models. All studies reported randomization, though none implemented allocation concealment or blinding of outcome assessment—two key measures often underreported in preclinical animal research. A statement of temperature control was present in 13 studies (41.9%), while avoidance of anesthetics with intrinsic neuroprotective properties was not explicitly addressed in any case. Only 6 studies (19.4%) used animals with relevant comorbidities such as diabetes or hypertension, potentially limiting clinical translatability. Sample size calculations were reported in 2 studies (6.5%), potentially reducing statistical power. All studies reported compliance with animal welfare regulations, and 28 (90.3%) disclosed potential conflicts of interest. Clinically relevant animal models were used in 19 studies (61.3%), and 17 (54.8%) addressed treatment timing in relation to the clinical scenario. Dose–response investigations were conducted in 11 studies (35.5%), and 23 (74.2%) described appropriate statistical methods. These findings indicate moderate methodological quality overall. While key criteria such as peer review and welfare compliance were met, important elements like blinding, comorbidity modeling, and sample size calculations were often lacking, limiting transparency and clinical translatability.

### 3.4. Dealing with Missing Data

Seven studies did not provide data on both mean and SD for each group regarding wound closure percentage and were contacted by e-mail with no responses (22, 24, 26, 62, 69, 76, 98); they were all excluded from the analysis entirely. In line with our protocol, only studies with extractable quantitative outcome data were included to ensure robustness and reproducibility of effect size calculations.

### 3.5. Statistical Analysis and Pooling

In total, 31 studies were included in the meta-analysis. The results are presented in a forest plot illustrating effect size of each study and the estimated overall effect size regarding gold nanoparticle treatment versus placebo, with wound closure percentage as the measured outcome (Figure 2). The effect size measure used was Cohen’s d and weights for the studies were calculated using the inverse-variance method, accounting for both within- and between-study variance. Additionally, a funnel plot was generated to assess publication bias among the studies included (Figure 3). The plot visually represented the effect sizes (Cohen’s d) on the *x*-axis and the standard error on the *y*-axis. The resulting funnel plot exhibited some degree of asymmetry, indicating potential publication bias.

### 3.6. Meta-Analysis Results

A total of 31 studies reporting wound closure percentages were included in a meta-analysis using a random-effects model. The forest plot (Figure 2) illustrates the individual and pooled effect sizes. The pooled standardized mean difference (Cohen’s d) was 4.52 (95% CI: 3.61 to 5.43), indicating a large effect size. The overall test of effect was statistically significant (*p* < 0.001). Heterogeneity was substantial (I^2^ = 85.28%), supporting the use of a random-effects model.

The funnel plot (Figure 3) displays effect sizes (Cohen’s d) on the *x*-axis and standard errors on the *y*-axis for the studies included in the meta-analysis. The plot shows visual asymmetry, with a noticeable lack of studies in the bottom left quadrant, suggesting potential publication bias due to small studies with low precision and smaller effect sizes.

**Figure 3 nanomaterials-15-01213-f003:**
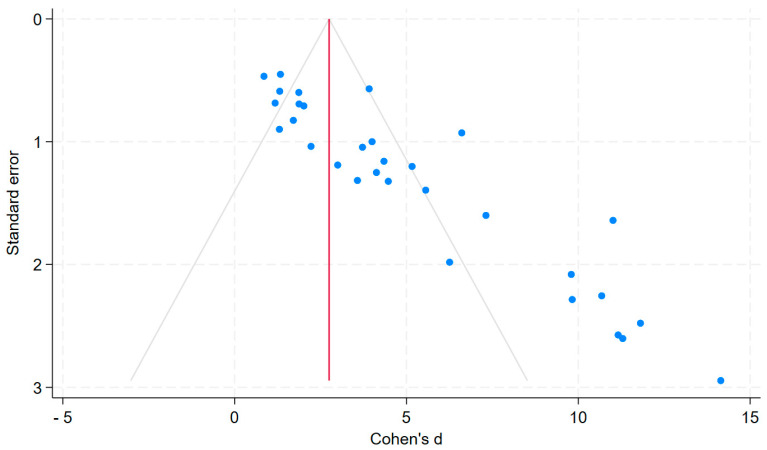
Funnel plot assessing publication bias.

## 4. Discussion

To our knowledge, this is the first systematic review and meta-analysis to quantitatively assess the effect of gold nanoparticle on wound healing in animal models. The meta-analysis showed an overall effect of gold nanoparticles on wound healing in animals compared to placebo.

We assessed publication bias using a funnel plot (Figure 3), which showed some asymmetry, mainly a lack of small studies with low precision and smaller effects. While we did not exclude negative findings, studies with non-significant results often lacked usable data. This pattern reflects a broader issue in preclinical literature, where underreporting negative results may skew the efficacy of interventions. As such, the pooled effect estimates should be interpreted with caution. The meta-analysis showed substantial heterogeneity, which may reflect differences in measurement time points, GNP formulations, and animal models. Although biological differences between wound models such as excisional, incisional, or infected wounds can influence healing outcomes [2,88], all included studies in this review employed full-thickness excisional wounds. Other wound types (e.g., incisional, partial-thickness, ischemic, or infected wounds) were excluded due to potential differences in healing mechanisms and outcome trajectories, which could confound the interpretation of pooled treatment effects. Some studies included animals with infection or diabetes, or evaluated antibacterial therapies, but the underlying model remained a full-thickness excisional wound. To account for variation in follow-up times across studies, we extracted wound closure data from the time point where the GNP intervention showed the largest effect compared to placebo. This approach was chosen to capture the peak efficacy of the treatment within each study and to avoid underestimation due to differing healing timelines. Peak effects were identified at various time points across studies, ranging from day 3 to 15. Despite minor variations in treatment protocols and formulations, the consistently positive outcomes suggest that GNPs may be an effective strategy for enhancing wound healing in full-thickness excisional injuries.

Most studies included in the meta-analysis reported the largest differences in wound closure between days 7 and 10. This likely reflects the transition from early-phase variability, driven by inflammation, into the proliferative phase characterized by re-epithelialization, angiogenesis, and collagen deposition [2,87]. By this time, acute inflammation has typically subsided, offering a more stable baseline to observe treatment effects [52,89].

GNPs were the primary therapeutic agent tested. Notably, most studies showing peak effects between days 7 and 10 involved topical application of GNP formulations. The highest therapeutic effects were observed in studies using spherical or ultrasmall GNPs delivered via hydrogels [61,84], nanofiber dressings [62], or functionalized protein composites in diabetic models [59]. The largest effect size reported used a low-dose spherical GNP gel formulation [68]. Over half the studies lacked comparable dosing data, limiting conclusions on dosage. These findings suggest that specific delivery forms, particularly hydrogels and nanofiber matrices may enhance the efficacy of GNP-based wound therapies. Despite differences in specific formulations and mechanisms, nearly all studies also reported effects on cell proliferation, migration, and/or angiogenesis. This convergence suggests that GNPs may broadly modulate key biological processes involved in wound repair.

All studies reported improved healing with GNPs. However, nanoparticle shapes and sizes varied widely—a critical factor, as these properties strongly influence biological activity [22,24,25]. Although this review did not stratify studies by nanoparticle size or shape, prior experimental work has shown that these physicochemical parameters, especially under systemic administration, significantly influence therapeutic outcomes in murine models [90]. While spherical nanoparticles seem to be the most common, several of the included studies utilized GNPs with less defined morphologies (rods, prisms, irregular shapes) [62,72,78]. This lack of standardization limits conclusions about structure–activity relationships. Exploring how size, shape, and dose influence efficacy remains an important direction for future research.

The preclinical exploration of GNPs exhibits promising potential in enhancing wound healing therapies [60,88]. However, several challenges must be addressed to translate these findings into effective clinical applications. Critical factors such as size, shape, and surface chemistry significantly influence the biocompatibility of GNPs, and their long-term safety is not fully understood [32,33,36]. These physical attributes affect cellular uptake, biodistribution, and toxicity, all of which are crucial parameters in wound healing [91,92]. Variations in synthesis methods lead to differences in biological activity and potential clinical efficacy [5]. Regulatory agencies require extensive safety and efficacy data, which are currently lacking for GNPs. The use of nanomaterials in humans raises ethical concerns regarding long-term exposure, environmental impact, and unforeseen adverse effects [93]. However, a factor to consider is the size of nanoparticles, which can be optimized to facilitate their recognition and clearance by the body’s immune system, specifically by macrophages [22,23]. While their small size facilitates systemic circulation, achieving targeted delivery to inflamed tissues or immune cells remains challenging [22]. Further research is needed to compare the efficacy and safety of GNPs under standardized experimental conditions, including long-term studies to assess chronic exposure effects, accumulation risks, and biodegradation over time [89]. Additionally, exploring encapsulation and controlled release through the integration GNPs with hydrogel matrices or liposomes may enhance drug delivery strategies [94]. The development of targeted therapies utilizing stimuli-responsive particles that activate specifically in diseased tissues could further improve therapeutic outcomes. Collaborative translational research among academia, industry, and regulatory bodies is essential to accelerate clinical development and establish safety standards by defining acceptable dosages, exposure limits, and safety protocols [23].

GNPs show great potential for the treatment of chronic wounds, but their use in clinical settings is still developing. Despite the promising findings from animal models, the clinical translation of GNP-based therapies remains uncertain. Only a minority of the studies included used models with comorbidities such as diabetes, which limits relevance to chronic wound conditions frequently encountered in humans [4]. Diabetic wound environments involve impaired angiogenesis, prolonged inflammation, and altered extracellular matrix remodeling—factors not fully replicated in healthy animal models. Consequently, efficacy in standard excisional models may overestimate real-world effectiveness [23]. Moreover, long-term safety data remains scarce. Gold nanoparticles can accumulate in tissues, and their clearance is size- and coating-dependent. Chronic exposure may lead to systemic adverse effects, especially under compromised physiological states like diabetes [5]. Because we only included studies with follow-up periods up to 15 days, we cannot draw conclusions about systemic toxicity or long-term biodistribution, which limits the assessment of clinical safety. Bridging the gap between short-term animal outcomes and real-world chronic wound settings will require standardized, long-term, comorbid models and human-relevant delivery platforms.

It is crucial to tackle issues related to biocompatibility, delivery methods, and regulatory approval through dedicated research and innovation to bring these therapies into practice. Future efforts should concentrate on optimizing particle design, performing comparative studies, and creating clear regulations to support safe and effective clinical applications [23].

## 5. Conclusions

Gold nanoparticles significantly improved wound healing outcomes in animal models compared to controls, with a large, pooled effect size and strong statistical support. Despite substantial heterogeneity, the use of a random-effects model allowed for a meaningful synthesis of results. These findings support the therapeutic potential of gold nanoparticles and underscore the need for further clinical investigation.

## Figures and Tables

**Figure 1 nanomaterials-15-01213-f001:**
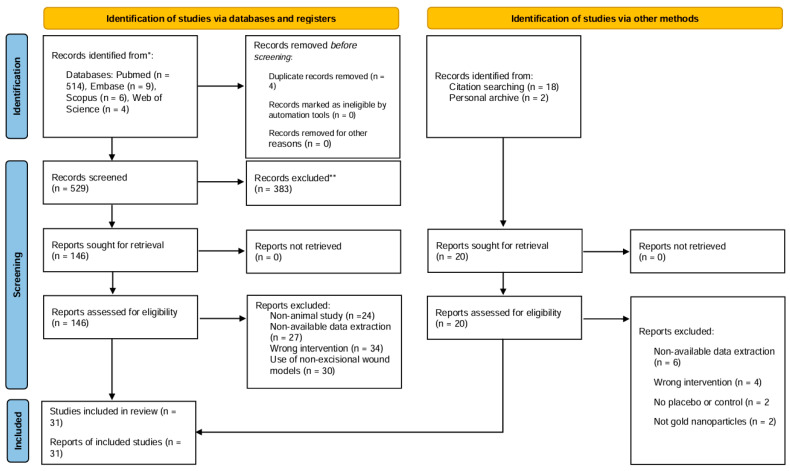
PRISMA flow diagram illustrating the study selection process for this systematic review. * Numbers of records identified from each database are shown in parentheses. ** Records excluded after title and abstract screening for not meeting inclusion criteria.

**Figure 2 nanomaterials-15-01213-f002:**
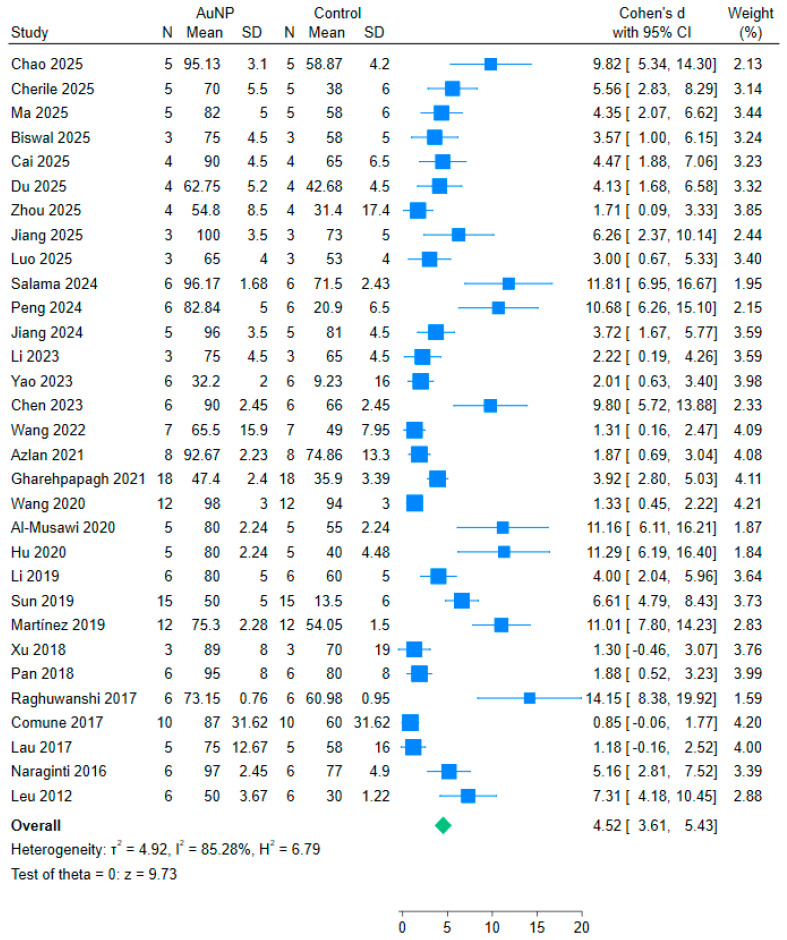
Forest plot of pooled effect sizes for wound closure. Data were extracted from published preclinical studies included in our systematic review ([7,20,40,56,57,58,59,60,61,62,63,65,66,67,68,69,70,71,72,73,74,75,76,77,78,79,80,81,82,83,84]).

**Table 2 nanomaterials-15-01213-t002:** Risk of bias assessment of the included studies based on the Cochrane Collaboration’s tool.Risk of bias was assessed across seven domains: random sequence generation (randomization), allocation concealment, blinding of participants and personnel, blinding of outcome assessment, incomplete outcome data, selective reporting, and other bias.

ID	Randomization	Allocation	Blinding	Assessment	Data	Reporting	Other
1. Chao 2025 [76]	Low	Unclear	High	High	Low	Low	Low
2. Gerile 2025 [82]	Low	Unclear	High	High	Low	Low	Low
3. Ma 2025 [77]	Low	Unclear	High	High	Low	Low	Low
4. Biswal 2025 [70]	Low	Unclear	High	High	Low	Low	Low
5. Cai 2025 [69]	Low	Unclear	High	High	Low	Low	Low
6. Du 2025 [65]	Low	Unclear	High	High	Low	Low	Low
7. Zhou 2025 [73]	Low	Unclear	High	High	Low	Low	Low
8. Jiang 2025 [83]	Low	Unclear	High	High	Low	Low	Low
9. Luo 2025 [66]	Low	Unclear	High	High	Low	Low	Low
10. Salama 2024 [84]	Low	Unclear	High	High	Low	Low	Low
11. Peng 2024 [71]	Low	Unclear	High	High	Low	Low	Low
12. Jiang 2024 [79]	Low	Unclear	High	High	Low	Low	Low
13. Li 2023 [81]	Low	Unclear	High	High	Low	Low	Low
14. Yao 2023 [75]	Low	Unclear	High	High	Low	Low	Low
15. Chen 2023 [20]	Low	Unclear	High	High	Low	Low	Low
16. Wang 2022 [64]	Low	Unclear	High	High	Low	Low	Low
17. Azlan 2021 [63]	Low	Unclear	High	High	Low	Low	Low
18. Gharehpapagh 2021 [67]	Low	Unclear	High	High	Low	Low	Low
19. Wang 2020 [60]	Low	Unclear	High	High	Low	Low	Low
20. Al-Musawi 2020 [62]	Low	Unclear	High	High	Low	Low	Low
21. Hu 2020 [61]	Low	Unclear	High	High	Low	Low	Low
22. Li 2019 [74]	Low	Unclear	High	High	Low	Low	Low
23. Sun 2019 [72]	Low	Unclear	High	High	Low	Low	Low
24. Martínez 2019 [59]	Low	Unclear	High	High	Low	Low	Low
25. Xu 2018 [80]	Low	Unclear	High	High	Low	Low	Low
26. Pan 2018 [58]	Low	Unclear	High	High	Low	Low	Low
27. Raghuwanshi 2017 [68]	Low	Unclear	High	High	Low	Low	Low
28. Comune 2017 [57]	Low	Unclear	High	High	Low	Low	Low
29. Lau 2017 [7]	Low	Unclear	High	High	Low	Low	Low
30. Naraginti 2016 [56]	Low	Unclear	High	High	Low	Low	Low
31. Leu 2012 [78]	Low	Unclear	High	High	Low	Low	Low

**Table 3 nanomaterials-15-01213-t003:** Characteristics of included studies. Studies fulfilling the criteria of: (1) peer-reviewed publication; (2) control of temperature; (3) random allocation to treatment or control; (4) blinded induction of model; (5) blinded assessment of outcome; (6) use of anesthetic without significant intrinsic neuroprotective activity; (7) use of animals with relevant comorbidities (e.g., aged, diabetic, hypertensive); (8) sample size calculation; (9) compliance with animal welfare regulations; (10) statement of potential conflicts of interest; (11) use of an appropriate animal model; (12) investigation of dose–response relationship; (13) timing of treatment relevant to clinical scenario; and (14) appropriate and clearly described statistical methods.

Publication	Year	(1)	(2)	(3)	(4)	(5)	(6)	(7)	(8)	(9)	(10)	(11)	(12)	(13)	(14)	Score
Chao [76]	2025	X		X			X			X	X	X			X	7
Gerile [82]	2025	X		X			X			X	X	X			X	7
Ma [77]	2025	X	X			X				X	X				X	6
Biswal [70]	2025	X	X	X		X				X					X	6
Cai [69]	2025	X	X	X		X				X	X	X			X	8
Du [65]	2025	X		X		X	X			X	X				X	7
Zhou [73]	2025	X	X			X				X					X	5
Jiang [83]	2025	X		X		X	X			X	X				X	7
Luo [66]	2025	X		X		X	X			X	X				X	7
Salama [84]	2024	X		X		X	X			X	X				X	7
Peng [71]	2024	X		X		X	X			X	X				X	7
Jiang [79]	2024	X	X	X		X	X	X		X		X	X		X	10
Li [81]	2023	X	X	X		X	X	X		X		X	X		X	10
Yao [75]	2023	X		X		X	X			X	X				X	7
Chen [20]	2023	X	X	X		X	X			X	X				X	8
Wang [64]	2022	X	X	X		X	X			X	X		X	X	X	10
Azlan [63]	2021	X		X		X	X			X			X	X	X	8
Gharehpapagh [67]	2021	X		X		X	X			X	X		X	X	X	9
Wang [60]	2020	X	X	X		X	X			X	X		X	X	X	10
Al-Musawi [62]	2020	X		X		X	X			X	X				X	7
Hu [61]	2020	X		X		X	X			X	X				X	7
Li [74]	2019	X	X	X		X	X	X		X	X				X	9
Sun [72]	2019	X		X		X	X			X	X				X	7
Martínez [59]	2019	X	X	X		X	X			X	X		X	X	X	10
Xu [80]	2018	X	X	X		X	X			X	X		X	X	X	10
Pan [58]	2018	X	X	X		X	X			X	X		X	X	X	10
Raghuwanshi [68]	2017	X	X	X		X	X			X	X		X	X	X	10
Comune [57]	2017	X	X	X		X	X			X	X		X	X	X	10
Lau [7]	2017	X		X		X	X			X	X		X	X	X	9
Naraginti [56]	2016	X	X	X		X	X			X	X		X	X	X	10
Leu [78]	2012	X	X	X		X	X			X	X		X	X	X	10

## Data Availability

The data supporting the findings of this study are available from the corresponding author upon reasonable request.

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
