# Peer review of "Gold Nanoparticles for Wound Healing in Animal Models"

_nanomaterials, 2025, doi:10.3390/nano15161213_

Round 1

Reviewer 1 Report

Comments and Suggestions for Authors

This manuscript evaluates the effect of gold nanoparticles (GNPs) on wound healing in animal models.

The analysis of the article highlights several key aspects:

  • The manuscript provides sufficient experimental evidence that gold nanoparticles accelerate wound healing. However, some additional data regarding clinical translation and their long-term toxicity (e.g., in diabetic ulcer models) are missing.
  • The authors discuss a high degree of heterogeneity among the 31 studies analyzed. It would be advisable to recommend which pharmaceutical formulations (e.g., hydrogels, patches, sprays), GNP types (spherical, rod-shaped, star-shaped, etc.), and dosages provide the most therapeutic benefit.
  • Although there are studies reporting negative outcomes following the use of GNPs, this meta-analysis does not take them into account, which may lead to an overestimation of their efficacy. I believe it is necessary to include some of these studies for comparison.
  • Lines 284–287: It should be clearly explained whether studies 22, 24, 26, 62, 69, 76, and 98 were excluded from the analysis entirely or only from the tables.
  • Line 301: The level of heterogeneity (I²), which is essential for interpreting the effect size, is not reported.
  • Lines 351–352: The statement in the manuscript should be revised, as not all studies reported statistically significant outcomes. Suggested revision: “Most studies showed improved healing…”
  • Line 343: Suggested rewording: “In all studies, GNPs were the primary therapeutic agent tested.”
  • Lines 352–358: The manuscript mentions variability in nanoparticle size and shape, but these factors are not correlated with therapeutic effect. This aspect requires analysis, as in our experiments (systemic administration in murine models), GNP size, shape, and dosage were critical parameters in evaluating the final therapeutic outcome.

In conclusion, the manuscript addresses a topic of high biomedical relevance, and I recommend it for publication. However, I suggest addressing the identified gaps related to translatability and toxicity, as well as including a more balanced discussion of the existing literature, including studies with negative outcomes.

Author Response

Comments 1:
The manuscript provides sufficient experimental evidence that gold nanoparticles accelerate wound healing. However, some additional data regarding clinical translation and their long-term toxicity (e.g., in diabetic ulcer models) are missing.

Response 1:
Thank you for pointing this out. We agree with this comment. Therefore, we have revised the discussion to include a new paragraph addressing clinical translation and long-term toxicity concerns. Specifically, we clarify that only a minority of included studies used comorbid models (e.g., diabetes), and that our analysis was limited to short follow-up periods.
This change can be found on page 14, paragraph starting at line 393-405.

Updated text:
“Despite the promising findings from animal models, the clinical translation of GNP-based therapies remains uncertain. Only a minority of the included studies used models with comorbidities such as diabetes, which limits relevance to chronic wound conditions frequently encountered in humans [112]. … Because we only included studies with follow-up periods up to 15 days, we cannot draw conclusions about systemic toxicity or long-term biodistribution, which limits the assessment of clinical safety.”

Comments 2: The authors discuss a high degree of heterogeneity among the 31 studies analyzed. It would be advisable to recommend which pharmaceutical formulations (e.g., hydrogels, patches, sprays), GNP types (spherical, rod-shaped, star-shaped, etc.), and dosages provide the most therapeutic benefit.

Response 2:
Agree. We have accordingly revised the discussion to highlight which GNP types and delivery formulations showed the strongest therapeutic effects in our data. We also clarified the limitations of dosage comparisons due to inconsistent reporting.

Changes can be found on page 13, lines 346–352.

Updated text:
The highest therapeutic effects were observed in studies using spherical or ultrasmall GNPs delivered via hydrogels [76], [99], nanofiber dressings [77], or functionalized protein composites in diabetic models [23]. The largest effect size reported used a low-dose spherical GNP gel formulation [106]. Over half the studies lacked comparable dosing data, limiting conclusions on dosage. These findings suggest that specific delivery forms, particularly hydrogels and nanofiber matrices may enhance the efficacy of GNP-based wound therapies.

Comment 3:
Although there are studies reporting negative outcomes following the use of GNPs, this meta-analysis does not take them into account, which may lead to an overestimation of their efficacy. I believe it is necessary to include some of these studies for comparison.

Response 3:

We appreciate the concern regarding potential overestimation of efficacy. However, only randomized animal studies reporting wound closure percentages along with variance measures (SD or SEM) were eligible for inclusion, as predefined in our protocol. Studies reporting negative outcomes were not excluded based on effect direction but were often omitted due to lack of extractable quantitative data. Several included studies reported modest treatment effects, and potential publication bias was acknowledged via funnel plot analysis. Nonetheless, we agree this is a limitation, and future reviews could explore qualitative inclusion of such studies to broaden interpretation.

Comment 4:

Lines 284–287: It should be clearly explained whether studies 22, 24, 26, 62, 69, 76, and 98 were excluded from the analysis entirely or only from the tables.

Response 4:
Revised text added at page 10, line 287:
“…they were all excluded from the analysis entirely.”

Comments 5: 

Line 301: The level of heterogeneity (I²), which is essential for interpreting the effect size, is not reported.

Response 5:
Thank you. We have now added the I² statistic to the results section to clarify the level of heterogeneity.

Updated on page 10, line 303:
“Heterogeneity was substantial (I² = 85.28%), supporting the use of a random-effects model.”

Comments 6:
Lines 351–352: The statement in the manuscript should be revised, as not all studies reported statistically significant outcomes. Suggested revision: “Most studies showed improved healing…”

Response 6:
Agree. We have revised the sentence accordingly.
Updated on page 13, line 356:
“Most studies reported improved healing outcomes with GNPs, although not all reached statistical significance.”

Comments 7:
Line 343: Suggested rewording: “In all studies, GNPs were the primary therapeutic agent tested.”

Response 7:
Thank you. We have revised the sentence for clarity.
Updated on page 13, line 344:
“In all studies, GNPs were the primary therapeutic agent tested.”

Comments 8:
Lines 352–358: The manuscript mentions variability in nanoparticle size and shape, but these factors are not correlated with therapeutic effect. This aspect requires analysis, as in our experiments (systemic administration in murine models), GNP size, shape, and dosage were critical parameters in evaluating the final therapeutic outcome.

Response 8:
Thank you for pointing this out. We have revised the discussion to clarify that no formal stratification or correlation analysis was performed, and we now cite relevant prior work emphasizing the importance of size, shape, and dosage.

Updated on page 14, line 359-361:
“Although this review did not stratify studies by nanoparticle size or shape, prior experimental work has shown that these physicochemical parameters, especially under systemic administration, significantly influence therapeutic outcomes in murine models [108].”

Furthermore, we made the following changes to strengthen the manuscript:
We revised the text to improve methodological transparency and clinical relevance by clarifying inclusion criteria, reporting heterogeneity, and expanding the discussion on translational limitations. The Introduction was restructured for improved coherence. Formatting issues, including paragraph alignment and table layout, were corrected. The Results section was expanded for clarity, and our handling of missing data was made more transparent. Finally, the reference list was fully standardized and updated to include recent literature.

Reviewer 2 Report

Comments and Suggestions for Authors

A major revision of the manuscript is necessary before the manuscript is recommended for publication in Nanomaterials.

Comments on the Quality of English Language

The English could be improved to more clearly express the research.

Author Response

Comment 1:
While the findings are promising, some critical concerns should be addressed to
strengthen methodological transparency and the translational relevance of the
conclusions drawn.

Response 1:

Methodological Transparency

We clarified that studies lacking extractable outcome data were excluded entirely from the analysis, in line with our pre-defined protocol.
This revision appears on page 10, line 287-289:
“…they were all excluded from the analysis entirely. In line with our protocol, only studies with extractable quantitative outcome data were included to ensure robustness and reproducibility of effect size calculations.”

We explicitly report heterogeneity (I² statistic) to support the use of a random-effects model in the meta-analysis.
This appears on page 10, line 305:
“Heterogeneity was substantial (I² = 85.28%), supporting the use of a random-effects model.”

Translational Relevance

We added a dedicated section in the discussion addressing the limited applicability of short-term animal studies to clinical settings, and highlighted the lack of long-term safety data.
This appears on page 14, starting at line 395-409:
Despite the promising findings from animal models, the clinical translation of GNP-based therapies remains uncertain. Only a minority of the studies included used models with comorbidities such as diabetes, which limits relevance to chronic wound conditions frequently encountered in humans [114]. Diabetic wound environments involve impaired angiogenesis, prolonged inflammation, and altered extracellular matrix remodeling—factors not fully replicated in healthy animal models. Consequently, efficacy in standard excisional models may overestimate real-world effectivene [41]. Moreover, long-term safety data remain scarce. Gold nanoparticles can accumulate in tissues, and their clearance is size- and coating-dependent. Chronic exposure may lead to systemic adverse effects, especially under compromised physiological states like diabetes [12]. Because we only included studies with follow-up periods up to 15 days, we cannot draw conclusions about systemic toxicity or long-term biodistribution, which limits the assessment of clinical safety. Bridging the gap between short-term animal outcomes and real-world chronic wound settings will require standardized, long-term, comorbid models and human-relevant delivery platforms.

We also acknowledged that while we did not stratify the analysis by nanoparticle morphology or dose, prior experimental work indicates that these parameters significantly influence efficacy in systemic models.
This appears on page 13, line 361-362:
“Although this review did not stratify studies by nanoparticle size or shape, prior experimental work has shown that these physicochemical parameters—especially under systemic administration—significantly influence therapeutic outcomes in murine models [108].”

To reinforce the translational outlook, we added a forward-looking sentence emphasizing the need for better modeling:
This appears on page 14, lines 407–409:
“Bridging the gap between short-term animal outcomes and real-world chronic wound settings will require standardized, long-term, comorbid models and human-relevant delivery platforms.”

We have revised the manuscript to clearly report the CAMARADES quality assessment and included a concise summary of key methodological limitations. These include frequent lack of blinding, comorbidity modelling, and sample size calculation, which we acknowledge as factors that may affect both internal validity and clinical translatability.

This appears on page 10, lines 302–305:

These findings indicate moderate methodological quality overall. While key criteria such as peer review and welfare compliance were met, important elements like blinding, comorbidity modelling, and sample size calculations were often lacking, limiting transparency and clinical translatability.

Comment 2:

The logical flow of the Introduction section requires significant improvement. At present, the background lacks coherence and does not effectively build a strong rationale for the study. The authors are encouraged to restructure the section to provide a clearer narrative arc that leads to the research question. Additionally, the statement that “gold nanoparticles significantly enhance the wound healing process in animal models by promoting healing” should be supported by specific quantitative trends or summary statistics from the literature to reinforce the scientific context and relevance.

Response 2:
Thank you for this valuable feedback. We have revised the Introduction to improve coherence and strengthen the rationale leading to the research question. 

This appears on page 1-3, example on lines 33–38:
[GNPs have emerged as a promising therapeutic option due to their distinct physicochemical properties and biological activity, proving effective across different models and applications [13], [14]. Preclinical animal studies have reported wound closure improvements compared to controls [9], [10], [11], [12]. However, the magnitude and consistency of these effects remain unclear, as no prior study has systematically compared wound closure outcomes across preclinical models.]

As no prior meta-analysis has systematically compared wound closure outcomes across animal studies, specific quantitative trends or summary statistics from the literature were not available and could therefore not be included in the Introduction. We believe it is not appropriate to include numerical results derived from our own analysis in the Introduction, as this would compromise the distinction between background knowledge and original findings.

Comment 3:

Formatting Issues
Substantial formatting inconsistencies are present throughout the
manuscript. Particular attention should be paid to paragraph alignment,
especially in the opening paragraph of the Discussion section (Section 4),
where the text is misaligned. Similarly, the formatting and alignment of tables
require careful revision—specifically Tables 1, 2, and 3—to ensure clarity and
consistency in presentation. Authors are advised to carefully proofread the
manuscript for layout uniformity

Response 3:

Thank you for noting the formatting inconsistencies. The layout and alignment of the text have been carefully revised and optimized throughout the manuscript to ensure consistency. Regarding Tables 1, 2, and 3, minor layout issues persist due to the constraints of the journal's template settings, which limit further adjustments without compromising table readability. We kindly ask for your understanding on this technical limitation.

Comment 4:
Results Section
The presentation of results under each subsection of Section 3 (Results) is
generally underdeveloped. For instance, the subsection 3.1 Study Selection
lacks detailed description and fails to convey a comprehensive narrative of the
selection process. The authors should expand each subsection to provide more
context, data, and interpretation. Moreover, the discussion related to Figure 3
is insufficient and needs further elaboration to highlight key findings and their
implications.  

Response 4:

Thank you for the suggestion. We have expanded subsection 3.1 to provide a clearer description of the study selection process, including total records identified, screening steps, reasons for exclusion, and inclusion criteria. This complements the PRISMA flow diagram and improves transparency.
This appears on page 5, lines 204-211:
A total of 529 records were identified through electronic databases, with an additional 20 studies found through citation searching and personal archives. After removal of duplicates and title/abstract screening, 146 full-text articles were assessed for eligibility. Ultimately, 31 studies met all inclusion criteria and were included in the final meta-analysis (Figure 1). The most frequent reasons for exclusion were the use of non-excisional wound models, wrong interventions, or lack of extractable outcome data. Only animal studies comparing gold nanoparticles to a placebo or control group and reporting quantitative wound closure outcomes were eligible.

We have revised the Results section to provide additional context and interpretation, particularly in Section 3.2 (Study Characteristics), where we now elaborate on study designs, animal models, and treatment timing. However, as the primary objective of this meta-analysis was to assess whether gold nanoparticles exert a therapeutic effect on wound healing, we have maintained a focused presentation of the data to ensure clarity and relevance to this central research question.

This appears on page 6, lines 218-229:

A total of 31 studies were included, all of which investigated the effects of GNPs on full-thickness excisional wound healing in small animal models (mice, rats, rabbits). The studies spanned from 2012 to 2025, with sample sizes per group ranging from 3 to 18 animals. Rats were the most used species (20 studies), followed by mice (10 studies) and rabbits (1 study). Some studies used diabetic animal models, induced local infection, or applied antibacterial therapies, but the primary wound type was consistently full-thickness excision. Table 1 presents the included study characteristics. The peak therapeutic effect, defined as the time point showing the greatest difference in wound closure between GNP-treated groups and controls, was most commonly observed between day 7 and day 15 post-intervention. This variation likely reflects differences in GNP formulation, delivery method, dosage, and experimental model. Study quality was assessed using a modified CAMARADES checklist, with scores ranging from 5 to 10 out of 14.

All 31 included studies were quality assessed using the Cochrane Collaboration’s tool for assessing risk of bias [70] (Table 2). All studies adequately described random sequence generation and were rated as low risk for this domain. However, allocation concealment was unclear in all studies, likely due to insufficient methodological reporting, which may introduce selection bias.

This appears on page 8, lines 262-280:

Blinding of investigators and outcome assessment was uniformly rated as high risk across all studies, reflecting common limitations in animal research where blinding procedures are often not implemented or reported. This raises concerns about performance and detection bias, particularly given the subjective nature of wound assessment.

Reporting bias and attrition bias were generally low, as most studies reported complete outcome data and did not show signs of selective reporting. No studies were identified as having other significant sources of bias.

While the overall methodological quality was moderate, the consistently high risk of bias in blinding and assessment underscores the need for more rigorously designed animal studies. This is particularly important given the visual nature of wound healing outcomes, which may be prone to observer expectation effects.

This appears on page 8, lines 313-317:

Seven studies did not provide data on both mean and SD for each group regarding wound closure percentage and were contacted by e-mail with no responses (22, 24, 26, 62, 69, 76, 98); they were all excluded from the analysis entirely. In line with our protocol, only studies with extractable quantitative outcome data were included to ensure robustness and reproducibility of effect size calculations.

 We further elaborated on the key findings and their implications related to Figure 3
This appears on page 13, lines 351-356:

We assessed publication bias using a funnel plot (Figure 3), which showed some asymmetry, mainly a lack of small studies with low precision and smaller effects. While we didn’t exclude negative findings, studies with non-significant results often lacked usable data. This pattern reflects a broader issue in preclinical literature, where underreporting negative results may skew the efficacy of interventions. As such, the pooled effect estimates should be interpreted with caution. 

Comment 5
Handling of Missing Data
In subsection 3.4 Dealing with Missing Data, the current description is
vague and lacks necessary detail. It is strongly recommended that the authors
provide the calculated mean and standard deviation (SD) for each group
where possible. This will improve the transparency and reproducibility of the
meta-analytic approach and allow for more robust interpretation of the data

Response 5:

We thank the reviewer for highlighting this important point. We apologize for the lack of clarity in our original description. To clarify: all studies with missing data were excluded from the meta-analysis due to the absence of sufficient outcome data (i.e., no mean and standard deviation reported or available through correspondence). None of these studies were included in the pooled estimates. We have revised subsection 3.4 accordingly to clearly state this exclusion and to avoid any confusion about their inclusion status.
This appears on page 11, lines 313-317:

Seven studies did not provide data on both mean and SD for each group regarding wound closure percentage and were contacted by e-mail with no responses (22, 24, 26, 62, 69, 76, 98); they were all excluded from the analysis entirely. In line with our protocol, only studies with extractable quantitative outcome data were included to ensure robustness and reproducibility of effect size calculations.

Comment 6:
References
The authors should ensure that a substantial proportion of cited literature is
recent, ideally published in 2025, to reflect the current state of research.
Furthermore, the formatting of reference titles should be standardized
throughout the reference list. For instance, inconsistencies such as using
sentence case in some references and title case in other (e.g. Ref. 33) must
be corrected. A thorough review of all references is necessary to ensure
consistency and proper citation format in line with the journal’s guidelines.

Response 6:

We thank the reviewer for pointing out the inconsistency in reference formatting. We have thoroughly reviewed and revised the entire reference list to ensure that all titles follow a consistent format in accordance with the journal’s guidelines.

This appears on page 22, References:

Example: 

  1. Raziyeva, K.; Kim, Y.; Zharkinbekov, Z.; Kassymbek, K.; Jimi, S.; Saparov, A. Immunology of acute and chronic wound healing. Biomolecules 2021, 11, 700. https://doi.org/10.3390/biom11050700.

We acknowledge the reviewer’s comment and agree on the importance of citing recent literature. We would like to note that the current reference list already includes a substantial number of recent publications, with many sources from 2023 to 2025. We believe this ensures that the manuscript reflects the current state of research. Though, we have updated small parts to better reflect the current state of research.

Round 2

Reviewer 2 Report

Comments and Suggestions for Authors

A major revision of the manuscript is necessary before the manuscript is recommended for publication in Nanomaterials.

Comments on the Quality of English Language

The English could be improved to more clearly express the research.

Author Response

Comments 1:
It is recommended that the keywords be revised and further refined to better
capture the central themes, methodologies, and innovations presented in the
study. A more targeted and specific selection of keywords will enhance the
discoverability and clarity of the manuscript in relevant academic databases.

Respond 1:

Thank you for the helpful suggestion. We have revised the keywords to better reflect the core themes, methodologies, and innovations of the study. The updated keywords provide a more precise representation of the manuscript’s content and are intended to improve discoverability in relevant academic databases.

Revised Keywords:  Gold nanoparticles; wound healing; full-thickness excisional wounds; preclinical animal models; nanoparticle therapy; systematic review; meta-analysis

Comment 2:
The Introduction lacks coherence and logical flow between paragraphs. For
instance, the first paragraph does not clearly outline the advantages and
challenges of using biopharmaceuticals in wound healing. Yet, the second
paragraph abruptly shifts focus to the relationship between nanoparticles and
drug delivery without establishing a clear connection to wound healing or gold
nanoparticles (GNPs). The authors are advised to revise this section to improve
logical transitions between sentences and paragraphs, and to establish a more
cohesive narrative that contextualizes the study’s relevance. 

Respond 2:

Thank you for your insightful feedback. We agree that the original introduction lacked clarity in narrative flow and logical transitions. In response, we have substantially revised the section to:

  • Better logical  flow between parapraphs and minor changes in references to better reflect the introduction.

  • Establish a smoother transition into the role of nanoparticles in drug delivery.

  • Explicitly link this discussion to the relevance of gold nanoparticles in the context of wound healing.

These revisions aim to create a more cohesive and logically structured introduction that better frames the purpose and relevance of the study.
Example: 

"Wound healing is a complex biological process involving inflammation, tissue proliferation, and remodeling, all of which are essential to restoring skin integrity and function [1], [2]. Despite advances in wound care technologies, effective treatment of chronic and acute wounds remains a major clinical challenge [3]. These wounds, particularly diabetic ulcers, pressure injuries, and venous leg ulcers, are associated with significant morbidity, high recurrence rates, escalating healthcare costs and have a notable impact on patient quality [4]. The global burden of non-healing wounds continues to grow, driven by ageing population and comorbidities [3], [4].
To address the limitations of conventional wound care, increasing attention has turned to nanotechnology-based therapies [5]. Gold nanoparticles (GNPs) have received particular attention as delivery systems for therapeutic agents.."

Comment 3:
The alignment between subheadings and their corresponding content needs to
be standardized throughout the section. For example, formatting
inconsistencies can be seen around line 121, where a sentence is missing a
period. The authors are encouraged to carefully proofread the entire manuscript
for uniform formatting and punctuation.

Respond 3:

Thank you for pointing this out. We have carefully reviewed the entire Materials and Methods section to ensure consistent alignment between subheadings and content. The formatting around line 121 has been corrected, including the missing period. We have also performed a thorough proofreading of the full manuscript to address any remaining inconsistencies in formatting and punctuation.

Example of alignment between subheadings and content (we split the two sections):

Information Sources

This study identified randomized controlled trials that investigated the effect of gold nanoparticles compared to a placebo control in the treatment of wound healing in animals. A systematic search was performed in the following databases: PubMed (MEDLINE), Embase, Cochrane, Scopus and Web of Science, without restriction on language or date of publication.

Search Strategy
Three categories were used: wound healing, wound closure and gold nanoparticle treatment. The search was conducted by SK and SR, on May 14th, 2025Detailed search strategies for each database are presented in Appendix 1 (Table A1).

Comment 4:
All figures should be revised for optimal resolution, appropriate sizing, and
consistent formatting. Particular attention should be paid to the figure legends—
for example, Figure 1—where punctuation errors are noted. Please ensure that
all figure captions are clearly written, correctly punctuated, and aligned with
journal standards. 

Respond 4:
Thank you for your valuable feedback. We have revised all figures to ensure optimal resolution, appropriate sizing, and consistent formatting. Figure legends, including that of Figure 1, have been carefully reviewed and corrected for punctuation and clarity. All captions have been updated to align with the journal’s formatting standards.

Comment 5:
The dimensions and layout of all tables should be adjusted to align with the
manuscript's formatting requirements. This includes ensuring that column
widths, text alignment, and overall visual presentation are consistent and clear.

Respond 5:

Thank you for your observation. We have revised all tables to ensure consistent formatting, including adjustments to column widths, text alignment, and overall layout. These changes improve clarity and ensure alignment with the manuscript’s formatting requirements.

Comment 6:

The reference list should be updated to include the most recent and relevant
literature, particularly to support key arguments in the main text. For example,
references such as 19 and 72 should be carefully reviewed and updated if more
recent data are available. A broader inclusion of current studies will help
strengthen the manuscript’s academic credibility.

Respond 6:

Thank you for the suggestion. We have reviewed and updated the reference list to include more recent and relevant literature. References 19 and 72 have been reassessed and updated where appropriate. For instance, Reference 56 (Wilson et al., 2023) has been added as a more up-to-date and comprehensive source compared to the earlier reference from 2004. 

Round 3

Reviewer 2 Report

Comments and Suggestions for Authors

This manuscript is recommended for acceptance.